# Length–Weight Relationships, Growth Models of Two Croakers (*Pennahia macrocephalus* and *Atrobucca nibe*) off Taiwan and Growth Performance Indices of Related Species

Shu-Chiang Huang [1,†], Shui-Kai Chang [1,*], Chi-Chang Lai [2,†], Tzu-Lun Yuan [3], Jinn-Shing Weng [2] and Jia-Sin He [2]

1 Graduate Institute of Marine Affairs, National Sun Yat-sen University, Kaohsiung 804, Taiwan
2 Coastal and Offshore Resources Research Center, Fisheries Research Institute, Council of Agriculture, Kaohsiung 806, Taiwan
3 Department of Applied Mathematics, Tunghai University, Taichung 974, Taiwan
* Correspondence: skchang@faculty.nsysu.edu.tw; Tel.: +88-675-250-050
† These authors contributed equally to this work.

**Abstract:** Information on age and growth is essential to modern stock assessment and the development of management plans for fish resources. To provide quality otolith-based estimates of growth parameters, this study performed five types of analyses on the two important croakers that were under high fishing pressure in southwestern Taiwan: *Pennahia macrocephalus* (big-head pennah croaker) and *Atrobucca nibe* (blackmouth croaker): (1) Estimation of length–weight relationships (LWR) with discussion on the differences with previous studies; (2) validation of the periodicity of ring formation using edge analysis; (3) examination of three age determination methods (integral, quartile and back-calculation methods) and selection of the most appropriate one using a k-fold cross-validation simulation; (4) determination of the representative growth models from four candidate models using a multimodel inference approach; and, (5) compilation of growth parameters for all *Pennahia* and *Atrobucca* species published globally for reviewing the clusters of estimates using auximetric plots of logged growth parameters. The study observed that features of samples affected the LWR estimates. Edge analysis supported the growth rings were formed annually, and the cross-validation study supported the quartile method (age was determined as the number of opaque bands on otolith plus the quartile of the width of the marginal translucent band) provided more appropriate estimates of age. The multimodel inference approach suggested the von Bertalanffy growth model as the optimal model for *P. macrocephalus* and logistic growth model for *A. nibe*, with asymptotic lengths and relative growth rates of 18.0 cm TL and 0.789 year$^{-1}$ and 55.21 cm, 0.374 year$^{-1}$, respectively. Auximetric plots of global estimates showed a downward trend with clusters by species. Growth rates of the two species were higher than in previous studies using the same aging structure (otolith) and from similar locations conducted a decade ago, suggesting a possible effect of increased fishing pressure and the need to establish a management framework. This study adds updated information to the global literature and provides an overview of growth parameters for the two important croakers.

**Keywords:** otolith; age determination; growth curves; edge analysis; growth performance index; quartile of marginal growth band method; k-fold cross-validation; back-calculation

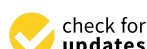



## 1. Introduction

Information on age and growth is essential to modern stock assessment and the development of management plans for fish resources; it has many functions such as converting input catch estimates from biomass to numbers and converting output numbers into biomass [1,2] and can affect the estimation of natural mortality and the mean age at maturity [3–5]. Establishing appropriate aging procedures and selecting representative growth models are thus important steps in developing stock assessments [1,6]. Even if the

growth curve has been estimated previously, reviewing the change in key parameters by a timely repeat of estimation is also beneficial to monitoring the stock status under fishing pressure and climate change [7–11].

Several approaches to estimate the growth curve have been used, including length-frequency analysis and growth bands analysis of fin rays, vertebrae, otoliths and scales. Otoliths are calcium carbonate structures formed inside the inner ears and grow with the deposition of continuous calcium carbonate layers, which can respond to both daily and seasonal changes. For teleost fishes, otoliths analysis is the most commonly used and reliable approach to age determination [12–14]. Age estimation using otoliths however requires high precision in laboratory techniques and estimating considerations, including validation of the periodicity of growth-rings formation, selection of age determination method (expression of age in integral count of growth rings or estimated value), and determination of representative growth curves.

Accuracy of age estimates is crucial in deriving age-based population parameters for making the right management decisions [15–17]. Edge analysis and marginal increment analysis (MIA) have been the most commonly used method to verify the increment periodicity of the otolith [18–21]. Edge analysis examines the timing of the translucent zone formation, and the ages are considered validated if only one opaque and one translucent band is formed annually [22–25]. The method is relatively easy to conduct compared to MIA, which plots the proportion of completion of the outermost increment against the month of capture to examine if it shows a yearly sinusoidal cycle [13,26,27] and can be used alone for the validation of annual periodicity of growth increment [18,24].

When determining the age under the microscope, several methods can be used for teleost fishes, including the commonly used integral counting method (integral method) where the age is the count of opaque bands, ignoring the growth within a year reflected in the width of the outmost translucent bands [28–31], and the back-calculate method where the age was determined as the proportion of the hypothesized hatching date to the sampling date plus the number of opaque bands [32,33]. Daily increments count has also been used [34]. Another method that can be explored is the quartile of marginal growth band method (quartile method) [35] to enhance the integral counting method by adding a quartile of the width of the marginal translucent band to the age. Ages estimates resulting from applying these methods are different and the accuracy needs to be investigated in advance.

The von Bertalanffy growth model (VBGM) has been widely used for fitting growth curves to the estimated age and the length data, but the standard VBGM is not always appropriate for all species since the ontogenetic changes in growth rates vary by species, and so alternative models should be explored [6,36]. The multimodel inference approach [37,38] is a useful way to select the most representative growth model from a suite of candidates for the species [27,39,40].

Plotting the estimated growth parameters together with those of other studies of the same species or related species can provide an overview of the similarities and differences between the current study and other global studies, and a chance to explore the differences [41–43]. Pauly and Munro [44] proposed a growth performance index based on logarithmized mean asymptotic length and relative growth rate obtained from the growth model, which allowed the identification of the bias in growth parameter estimates within a species while also revealing the difference between species. The auximetric plot created using these two parameters can help to visualize the clusters formed by the same species [43,45]. It has been applied widely to compare estimated growth parameters from different studies [27,46–48].

Sciaenidae includes 70 genera with around 282 species distributed in the Atlantic, Indian, and Pacific Oceans [49] and are important to coastal fisheries of many countries. There were 23 species in 12 genera of Sciaenidae recorded in Taiwan waters [50]. The high economic value of the Sciaenidae and its characteristics of mass migration and concentration during the breeding season make it easy to be caught in large quantities. Sciaenidae species

are an important target to the trawl fishery of Taiwan. Among them, *Pennahia macrocephalus* ([51]; big-head pennah croaker) and *Atrobucca nibe* ([52]; blackmouth croaker) are two of the most abundant commercial species in southwestern Taiwan (Figure 1). They are widely distributed across Indo-Pacific waters with depth between 20 to 200 m, feeds on small invertebrates and crustaceans. Total landing of *Pennahia* sp. once reached a peak of 8.6 thousand tons in 1996 and declined to 852 tons in 2015, while that of *A. nibe* was 2.5 thousand tons in 1993 and declined to less than 500 tons after 2011 [50]. Lack of logbook from coastal fisheries due to exemption of submission responsibility [53] has hindered the assessment of the stock using catch data; studies of biological parameters are considered as alternatives for providing supplemental information on the stock.

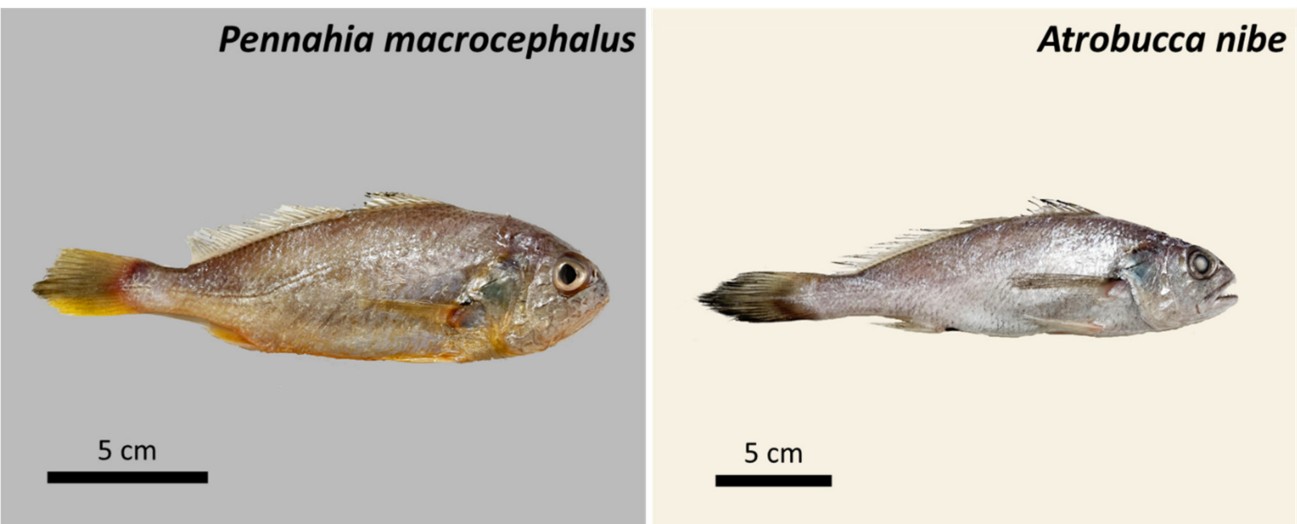

**Figure 1.** Two important croaker species occurred in Taiwan. *Pennahia macrocephalus* (big-head pennah croaker, age = 3.5) and *Atrobucca nibe* (blackmouth croaker, age = 2).

As indicated above, we suppose that different age determination methods may affect the results of growth estimates, the most optimal growth model of these two species may not be the VBGM, and same fish species may have a relatively similar growth performance [44]. Therefore, five types of analyses were performed on the two species in this study: (1) estimation of length–weight relationships (LWR) which is also a key element in the examination of fish biology and contains valuable information to assess the general health condition of fish species [54–56]; (2) validation of the periodicity of ring formation using edge analysis; (3) examination of three age determination methods and selection of the most appropriate one using a k-fold cross-validation simulation study [57,58]; (4) determination of the best fitting growth models from four candidate models; and, (5) compilation of growth parameters for all *Pennahia* and *Atrobucca* species published globally for reviewing the clusters (groups) of estimates using auximetric plots. Most growth studies on the two species were conducted a decade ago and the majority of them were published in grey literature or in Chinese. This study adds updated information to the global literature and provides an overview of growth parameters for the two important croakers.

## 2. Materials and Methods

### 2.1. Data Sources and Estimation of Length-Weight Relationships (LWR)

Biological data: total length (TL) in cm, body weight (BW) in g, sex, and otolith samples of *P. macrocephalus* in southwestern Taiwan waters were collected monthly from July to October 2018 and February to August 2021 in the fish markets of Zihquan and Tungkang (Figure 2). Biological data of *A. nibe* were collected monthly from July 2017 to August 2018 at the fish markets of, or from fishing boats based in, Zihquan, Tungkang, Fangliao and Liuqiu (Figure 2). Excluding samples abandoned due to damage caused during otolith

processing or disagreements on the number of ring marks between two age readings, a total of 359 *P. macrocephalus* and 386 *A. nibe* samples were used in this study.

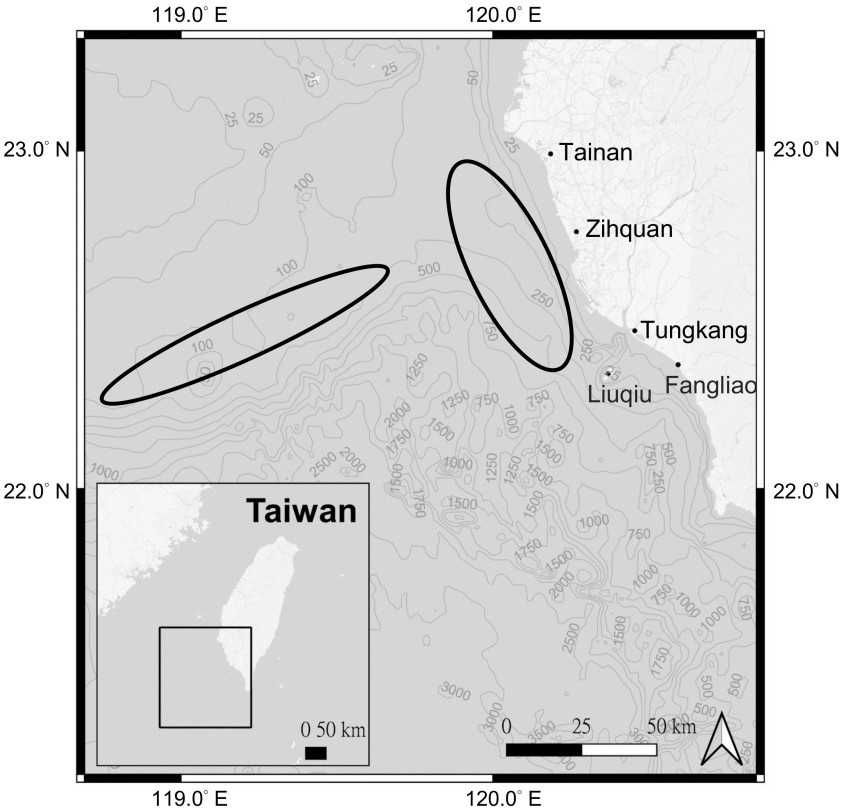

**Figure 2.** Black circles indicate the major fishing areas of *P. macrocephalus* and *A. nibe* from the trawlers based in Zihquan, Tungkang, Fangliao and other ports in southwestern Taiwan. The market samples used in this study came from these fishing areas.

The length–weight relationship (LWR) for each species was estimated using a power function of the form $BW = a \times TL^b$ where *a* is the coefficient of the power function, and *b* is the allometric coefficient. Parameters *a* and *b* were estimated with a linear regression of ln(*BW*) against ln(*TL*). Size distribution and LWR by sex for each species were also examined. The difference of the parameters from the linear regression fits between sexes for both species were tested using ANOVA [59].

*2.2. Otolith Processing, Periodicity of Increment and Age Determination*

Otoliths (sagittae, lapilli, and asterisci, Supplementary Figure S1) were extracted from the fish samples. Sagittae, the largest of the three otoliths, are commonly used for estimating fish age [12] and were used for this study. They were firstly washed with water and air-dried for at least 24 h. Next, the otoliths were prepared for transverse sections by embedding the whole otolith in epoxy resin and epoxy hardener mixture (with the ratio 10:0.95), placing them at 60 °C for 2 h, and sectioning transversely with a low-speed diamond wheel saw (approximately 500 μm thick). Finally, the otolith sections were polished with silicone abrasive grinding paper (1200 to 4000 grit) until the rings became clear. The otoliths were examined using a microscope with transmitted light, by which the periods of fast growth accreted otolith material are translucent whereas it is opaque during slow growth periods, which is different from the observation viewed under a dissecting microscope with reflected light [60]. A translucent zone and subsequent opaque zone are referred to as an 'increment'. Ring marks on the otoliths were counted and measured by at least two different readers or three times by the same reader. The results were accepted when the readings matched; otherwise, the results were abandoned and were not included in this study.

Edge analysis was used to examine the periodicity of increment formation (the timing of the translucent zone formation) and was calculated through the monthly proportion of the translucent zone on the outer edge of the sectioned otolith [23,24]. Subsamples for edge analysis were selected by month from the most abundant age groups, with efforts to restrict the number of age groups [13] (n = 296 for *P. macrocephalus* with age 2–4, TL 13.7–19.6 cm and 186 for *A. nibe* with age 4–6, TL 23.2–53.8 cm).

Three methods for determining the annual increments were conducted and evaluated for selecting the optimal one. (1) Integral counting method (integral method), where the opaque bands were counted from the nucleus to the edge of the otolith. This method was commonly used for many fish species, but since the marginal translucent band was ignored, this method may underestimate the actual age of the fish (Figure 3a). (2) Quartile of marginal growth band method (quartile method). The age was determined in the same way as the integral counting method when the edge of the otolith is an opaque band. If the edge of the otolith is a translucent band, the width of the marginal translucent band was compared to the width of the previous complete annulus to the approximate value of 0.25, 0.5 or 0.75; and the age was determined as the quartile plus the number of opaque bands (Figure 3b) [35]. (3) Back-calculation method. The age was determined as the proportion of the hypothesized hatching date to the sampling date plus the number of opaque bands [32].

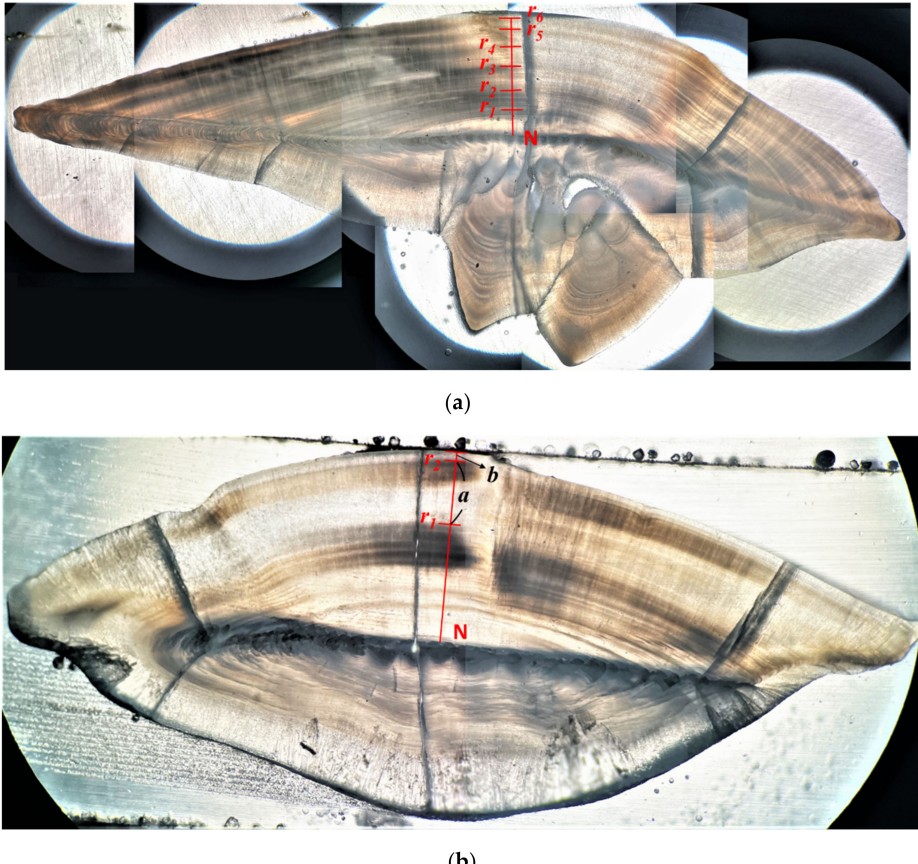

(a)

(b)

**Figure 3.** Sample images of sagittal otolith (transverse sections) showing the determination of ages by different methods. (**a**) Integral counting method (sample of *A. nibe*; total length 41.5 cm): $r_n$ indicate the ring marks on the outer margin of the opaque bands used for determining age class *n*. (N: nucleus). For this case, the age is six. (**b**) Quartile of marginal growth band method (quantile method, sample of *P. macrocephalus*; total length 16.8 cm): $r_1$ and $r_2$ indicate the ring marks on the outer margin of the opaque bands. *a* is the width of the most recent formed annulus, and *b* is the width of the hyaline band extended from the margin of the outmost opaque band to the edge. Ages were determined as the number of opaque bands plus approximate quartile value of b against a. For this case, the age is 2.25.

A k-fold cross-validation simulation study [57,58] was performed to evaluate the goodness of fit of the growth model to the age-length data estimated by each of the three methods. The most commonly used VBGM was used as the reference model. The age-length data, with length data and age estimates from the three methods, were firstly split into four groups based on the results of the integral counting method (e.g., ages 0–1, age 2, age 3, and ages 4–5). Data within each age group were then randomly divided into five sub-groups, or folds, of approximately equal size. Each fold contains data from all age groups. The first fold (across the four age groups) was treated as a validation set, and the VBGM was fit on the remaining four folds (training set). The mean square error (MSE) was calculated from the observed length of the validation set, and the predicted length was estimated from the VBGM developed using the training set. Afterwards, the other folds were treated as validation sets, and the VBGM-developing and MSE-estimation were repeated. This loop was repeated 100 times to finally obtain the mean and standard deviation (sd) of the MSE for each type of age determination method.

### 2.3. Growth Parameters Estimation

An information-theoretic, multi-model inference approach was used to determine the optimal growth model for both species [37,38]. The candidate growth models examined include non-sigmoidal VBGM and three sigmoidal models: Gompertz, Logistic, and Richards models [61–64]. The first three are the most common three-parameter models [65], and the last is a four-parameter model.

$$\text{Von Bertalanffy model}: \ L_t = L_\infty \left(1 - e^{-K(t-t_0)}\right) \tag{1}$$

$$\text{Gompertz model}: \ L_t = L_\infty e^{-e^{-K(t-t_0)}} \tag{2}$$

$$\text{Logistic model}: \ L_t = L_\infty \left(1 + e^{-K(t-t_0)}\right)^{-1} \tag{3}$$

$$\text{Richards model}: \ L_t = L_\infty \left(1 + \frac{1}{p}e^{-K(t-t_0)}\right)^{-p} \tag{4}$$

where $L_t$ is the length (TL in cm) at age $t$, $L_\infty$ is the mean asymptotic length, $p$ is a dimensionless parameter of Richards model, and $t_0$ is the theoretical age when length equals zero for the VBGM or the age at the inflection point of the growth curve for the other models. Parameter $k$ (year$^{-1}$) is a relative growth rate, describing how quickly the asymptotic length is approached or the rate of exponential decrease of the relative growth rate with age [37,66].

The length-at-age data (combined and sex-specific) were fitted to the four candidate models using non-linear least squares in R version 4.1.2 [67]. The small-sample bias-corrected form (*AICc*) of the AIC (Akaike Information Criterion) was used for model selection [65,68,69]. The model with the smallest *AIC* value ($AIC_{c,\,min}$) was selected as the 'best' among the models tested. *AIC* differences $\Delta = AIC_{c,\,min} - AIC_{c,\,i}$ were computed overall candidate models $i$. The Akaike weight, $w_i$, of each model was then calculated using these differences to quantify the plausibility, which is considered as the weight of evidence in favor of model $i$ being the best of the available set of models [37,69].

$$w_i = \frac{e^{-0.5\Delta_i}}{\sum_{i=1}^{4} e^{-0.5\Delta_i}} \tag{5}$$

### 2.4. Global Estimate Compilation

A literature (e.g., books, peer-reviewed articles and project reports) search was conducted on the estimation of growth parameters for *Pennahia* and *Atrobucca* species; however, only four *Pennahia* and two *Atrobucca* species (i.e., *P. macrocephalus*, *P. argentata*, *P. pawak*, *P. anea*, *A. nibe* and *A. alcocki*) were available. Global estimates of growth parameters of the six species were compiled from the literature in Table 1 and Supplementary Table S1.

Estimates of *k* were in year$^{-1}$; estimates of $L_\infty$ were in cm in either standard length (SL), fork length (FL), total length (TL), body length (BL) or not available (NA). Most of the $L_\infty$ were in TL; thus, the estimates were converted to TL using the formulae listed below [70–72] or assumed to be TL for the cases of NA.

$$TL = 0.651 + 1.162 \text{ SL } (R^2 = 0.997, \text{ for } P.\ macrocephalus)$$
$$TL = -0.37531 + 1.225 \text{ SL } (R^2 = 0.965, \text{ for } P.\ argentata) \tag{6}$$
$$TL = FL \text{ (For } P.\ anea)$$

**Table 1.** Global estimates of growth parameters of *Pennahia* and *Atrobucca* spp. from the literature. Lengths are in the unit of cm and *k* in year$^{-1}$. Refer to Table S1 for detailed information.

| Code | Species | Region | Material (Method) | $L_\infty$ (TL) | $k$ (cm y$^{-1}$) | $\varphi$ | Ref |
|---|---|---|---|---|---|---|---|
| m_1 | *P. macrocephalus* | Beibu Gulf, SCS | Otolith (I) | 27.371 | 0.408 | 2.485 | [29] |
| m_2 | *P. macrocephalus* | Beibu Gulf, SCS | Len. freq. | 28.899 | 0.520 | 2.638 | [70] |
| m_3 | *P. macrocephalus* | Beibu Gulf, SCS | Scale | 31.502 | 0.590 | 2.768 | [73] |
| m_4 | *P. macrocephalus* | Southwest, TW | Otolith (D) | 27.028 | 0.596 | 2.639 | [34] |
| m_5 | *P. macrocephalus* | Yunlin, TW | Otolith (B) | 21.358 | 0.371 | 2.228 | [32] |
| mc_1 | *P. macrocephalus* | Southwest, TW | Otolith (I) | 19.592 | 0.328 | 2.099 | This study |
| mc_2 | *P. macrocephalus* | Southwest, TW | Otolith (Q) | 18.004 | 0.789 | 2.408 | This study |
| mc_3 | *P. macrocephalus* | Southwest, TW | Otolith (B) | 18.258 | 0.617 | 2.313 | This study |
| a_1 | *P. argentata* | TW strait | Scale | 30.887 | 0.978 | 2.970 | [74] |
| a_2 | *P. argentata* | TW strait | Scale | 35.713 | 0.375 | 2.680 | [74] |
| a_3 | *P. argentata* | Southern, ECS | Scale | 28.878 | 1.207 | 3.003 | [74] |
| a_4 | *P. argentata* | Southern, ECS | Scale | 34.733 | 0.315 | 2.580 | [74] |
| a_5 | *P. argentata* | Beibu Gulf, SCS | Scale | 30.500 | 0.350 | 2.513 | [75] |
| a_6 | *P. argentata* | Beibu Gulf, SCS | Scale | 28.230 | 0.500 | 2.600 | [76] |
| a_7 | *P. argentata* | Northern, SCS | Len. freq. | 38.200 | 0.420 | 2.787 | [71] |
| a_8 | *P. argentata* | Beibu Gulf, SCS | Len. freq. | 31.500 | 0.350 | 2.541 | [71] |
| a_9 | *P. argentata* | TW strait | Otolith | 36.301 | 0.377 | 2.696 | [30] |
| a_10 | *P. argentata* | Southern Sea, SK | Otolith | 35.529 | 0.380 | 2.681 | [31] |
| a_11 | *P. argentata*♀ | Ariake Sound, JP | Otolith | 31.200 | 0.384 | 2.573 | [77] |
| a_12 | *P. argentata*♂ | Ariake Sound, JP | Otolith | 29.400 | 0.360 | 2.493 | [77] |
| a_13 | *P. argentata* ♀ | Tachibana Bay, JP | Otolith | 35.400 | 0.272 | 2.533 | [28] |
| a_14 | *P. argentata* ♂ | Tachibana Bay, JP | Otolith | 29.000 | 0.342 | 2.459 | [28] |
| a_15 | *P. argentata*♀ | Omura Bay, JP | Otolith | 37.000 | 0.256 | 2.545 | [28] |
| a_16 | *P. argentata*♂ | Omura Bay, JP | Otolith | 37.700 | 0.181 | 2.410 | [28] |
| a_17 | *P. argentata*♀ | The Sea of Goto, JP | Otolith | 44.500 | 0.241 | 2.679 | [28] |
| a_18 | *P. argentata*♂ | The Sea of Goto, JP | Otolith | 45.200 | 0.195 | 2.600 | [28] |
| a_19 | *P. argentata* | Seto Inland Sea | Scale | 39.56 | 0.307 | 5.68 | [78] |
| p_1 | *P. pawak* | Beibu Gulf, SCS | Scale (Logistic) | 22.030 | 0.580 | 2.449 | [79] |
| p_2 | *P. pawak* | Beibu Gulf, SCS | Len. freq. | 24.150 | 0.390 | 2.357 | [80] |
| p_3 | *P. pawak* | Beibu Gulf, SCS | Len. freq. | 22.050 | 0.320 | 2.192 | [80] |
| an_1 | *P. anea* | Paradeep, IN | Len. freq. | 30.300 | 0.860 | 2.897 | [81] |
| an_2 | *P. anea* | San Miguel Bay, PHI | Len. freq. | 20.000 | 0.600 | 2.380 | [82] |
| an_3 | *P. anea* | Manila Bay, PHI | Len. freq. | 26.500 | 1.400 | 2.993 | [83] |
| an_4 | *P. anea* | Penang/Perak, MA | NA | 34.200 | 0.400 | 2.670 | [84] |
| an_5 | *P. anea* | Bombay, IN | Len. freq. | 24.500 | 0.640 | 2.585 | [85] |
| an_6 | *P. anea* | Bombay, IN | Len. freq. | 26.000 | 1.200 | 2.909 | [86] |
| an_7 | *P. anea* | Rameswaram, IN | Len. freq. | 23.300 | 1.260 | 2.835 | [87] |
| an_8 | *P. anea* | Mandapam, IN | Len. freq. | 26.000 | 0.980 | 2.821 | [88] |
| an_9 | *P. anea* | Bombay, IN | Len. freq. | 27.300 | 1.940 | 3.160 | [89] |
| an_x | *P. anea* | Andhra Pradesh, IN | Len. freq. | 33.000 | 0.700 | 2.882 | [90] |
| an_y | *P. anea* | Indonesia | Len. freq. | 23.890 | 0.840 | 2.681 | [91] |
| n_1 | *A. nibe* | Formosa Strait | NA | 58.500 | 0.116 | 2.599 | [92] |
| n_2 | *A. nibe* | North, TW | Len. freq. | 58.700 | 0.145 | 2.699 | [92] |
| n_3 | *A. nibe* | North, TW | Len. freq. | 37.100 | 0.354 | 2.688 | [92] |
| n_4 | *A. nibe* | North, TW | Len. freq. | 37.100 | 0.523 | 2.857 | [92] |
| n_5 | *A. nibe* | North, TW | Len. freq. | 57.300 | 0.177 | 2.764 | [92] |
| n_6 | *A. nibe* | South, SK | Len. freq. | 50.900 | 0.227 | 2.769 | [92] |
| n_7 | *A. nibe* | South, SK | Len. freq. | 46.900 | 0.297 | 2.815 | [92] |
| n_8 | *A. nibe* | TW | Scale | 56.290 | 0.120 | 2.580 | [93] |
| n_9 | *A. nibe* | ECS | Scale | 43.200 | 0.252 | 2.672 | [94] |
| n_10 | *A. nibe* | Oman Sea | Otolith | 50.000 | 0.200 | 2.699 | [95] |
| n_11 | *A. nibe* | Guei-Shan Island, TW | Otolith | 48.060 | 0.275 | 2.803 | [96] |

**Table 1.** *Cont.*

| Code | Species | Region | Material (Method) | $L_\infty$ (TL) | $k$ (cm y$^{-1}$) | $\varphi$ | Ref |
|------|---------|--------|-------------------|-----------------|-------------------|-----------|-----|
| n_12 | *A. nibe* | Guei-Shan Island, TW | Otolith | 64.910 | 0.147 | 2.792 | [96] |
| n_13 | *A. nibe*♀ | NA | Len. freq. | 45.000 | 0.225 | 2.659 | [92] |
| n_14 | *A. nibe*♂ | NA | Len. freq. | 40.100 | 0.238 | 2.583 | [92] |
| n_15 | *A. nibe* ♀ | Southwest, TW | Otolith | 53.384 | 0.183 | 2.717 | [97] |
| n_16 | *A. nibe* ♂ | Southwest, TW | Otolith | 52.577 | 0.187 | 2.713 | [97] |
| n_17 | *A. nibe* ♀ | Northeast, TW | Otolith | 49.527 | 0.227 | 2.746 | [98] |
| n_18 | *A. nibe*♂ | Northeast, TW | Otolith | 48.417 | 0.201 | 2.673 | [98] |
| nc_1 | *A. nibe* | Southwest, TW | Otolith (VBGM) | 68.149 | 0.130 | −0.919 | This study |
| nc_2 | *A. nibe* | Southwest, TW | Otolith (Logistic) | 53.111 | 0.394 | 2.958 | This study |
| al | *A. alcocki* | Pakistani waters | Len. freq. | 47.250 | 0.180 | 2.604 | [99] |

Note: $L_\infty$ (TL), the $L_\infty$ value in total length (TL, might be converted from the original value). For Sex: M, male; F, female; C, sex-combined. For Region: SCS, the South China Sea; TW, Taiwan; ECS, the East China Sea; SK, South Korea; PHI, Philippines; MA, Malaysia; IN, India; JP, Japan. For method: I, integral counting method; Q, quartile method; B, back-calculated method; D, daily growth ring method. NA, information not available.

Growth parameters were estimated using different methods: the aging of otoliths (sagittae) or scales, length-frequency data (LFD), or unknown method. Growth parameters for the two species in this study were compiled into the global estimates table (62 estimates in total) for classification analysis. All the parameters were from VBGM, except for the records coded as p_1 and nc_2, which were Logistic.

The two variables, $\log(L_\infty)$ and $-\log(k)$, of the growth performance index formula, $\varphi = 2\log(L_\infty) + \log(k)$ [44], were used to identify the clusters of growth estimates from the global summary using the analysis of the auximetric plot [43,45,46]. Ellipses of 95% confidence intervals were also produced in the plot except for *P. pawak* with three records and *A. alcocki* with one record only.

## 3. Results

### 3.1. Samples and Length-Weight Relationships (LWR)

For *P. macrocephalus*, 359 individuals with total lengths and bodyweight ranging from 12.20 to 20.13 cm and from 21.70 to 104.82 g, respectively, were examined (Supplementary Table S2 with raw data and summary statistics). Males ranged from 12.20 to 20.13 cm TL and 21.70 to 104.70 g ($n = 207$) and females from 13.45 to 19.62 cm TL and 27.17 to 104.82 g ($n = 152$). The ANOVA test results indicated that there is no significant difference of the LWR parameters between sexes ($p = 0.095$, df = 355). Therefore, the sex combined LWR equation is presented as $BW = 1.122 \times 10^{-3} \cdot TL^{3.061}$ ($R^2 = 0.892$) (Figure 4) (or $BW = 2.770 \times 10^{-2} SL^{2.937}$). To examine if the coefficients have been biased by possible overweighting by more samples from the breeding season (June to September), an additional LWR estimation was performed using limited sub-samples: all data in a month were used when the monthly sample size was <30, or 30 fish were randomly selected when the monthly sample size was >30. The resulted average was $b = 3.122$ ($n = 10$), suggesting that reducing sample size from the breeding season does not reduce the estimation of $b$ and that the current estimates using all sample data are representative.

For *A. nibe*, 386 individuals with total lengths and bodyweight ranging from 17.02 to 56.6 cm and from 46.30 to 1941.60 g, respectively, were examined (Supplementary Table S3 with raw data and summary statistics). Males ranged from 17.01 to 52.70 cm TL and from 46.30 to 1235.07 g ($n = 152$) and females from 18.65 to 56.60 cm TL and from 78.37 to 1941.60 g ($n = 234$). The ANOVA test results indicated that there is no significant difference of the LWR parameters between sexes ($p = 0.156$, df = 382). The length–weight relationship equation is thus presented as $BW = 1.448 \times 10^{-2} \cdot TL^{2.886}$ ($R^2 = 0.985$) for sex combined.

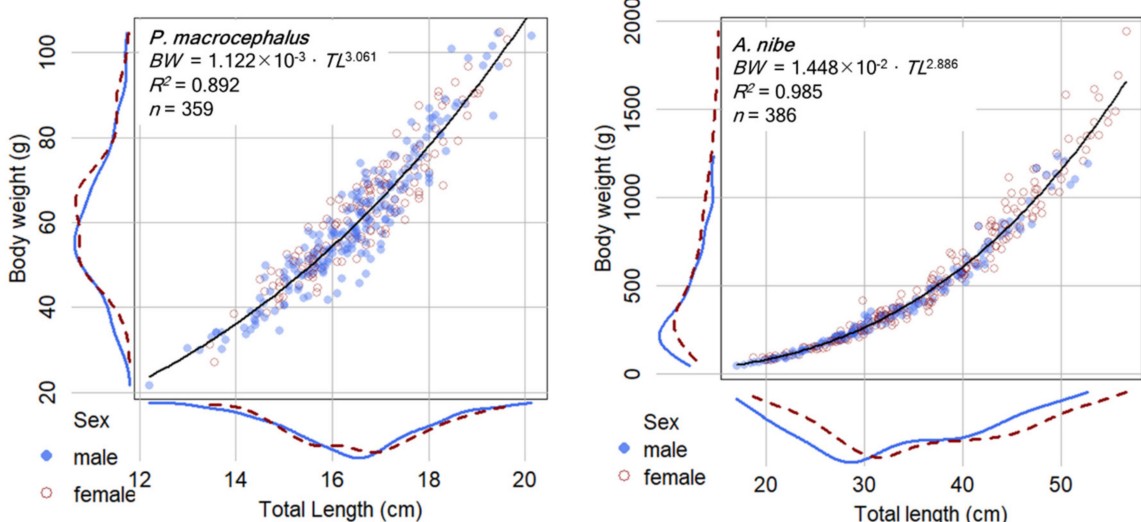

**Figure 4.** Total length (TL) and body weight (BW) data and the estimated length-weight equation of *P. macrocephalus* and *A. nibe*. The blue lines and circles are male, and the red lines and circles are female.

### 3.2. Periodicity of Increment Formation—Edge Analysis

Monthly percent occurrence of the translucent band at the edge for the two croakers is depicted in Figure 5. For *P. macrocephalus*, we were not able to obtain samples for November to January since it did not coincide with the active fishing season. However, the trend from the lowest proportion of translucent bands in February to the high level from June to October could still be observed. Although there was data deficiency during the winter period, the apparent trend suggested that the translucent bands started to form once a year near the end of winter and the early spring.

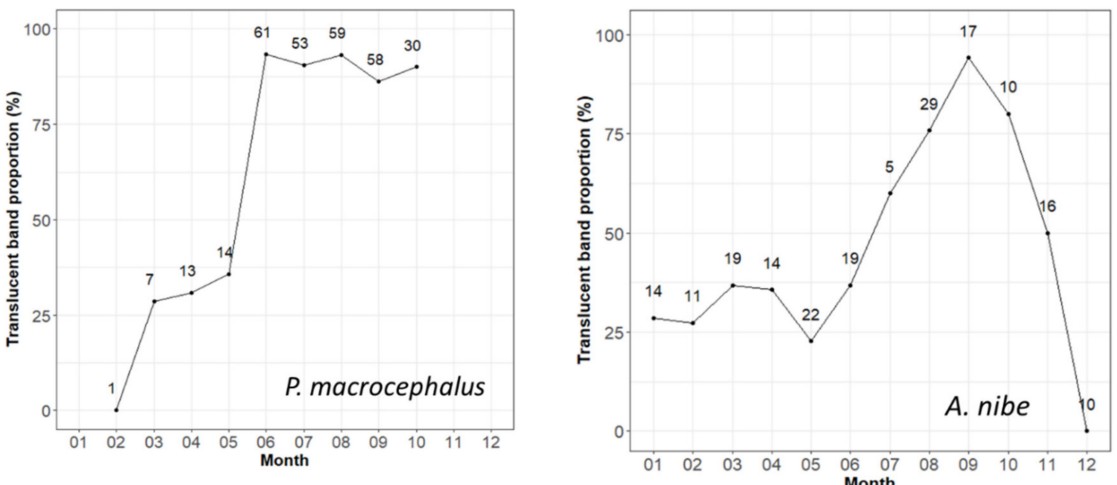

**Figure 5.** Monthly percent occurrence of the translucent band at the edge of samples from *P. macrocephalus* and *A. nibe*. Numbers indicate the sample sizes.

On the other hand, *A. nibe* can be caught throughout the year. The result of the edge analysis showed that the proportion of translucent bands started to increase from January to September (100%), with a drop in May (likely caused by sampling issues) followed by a sharp decline to 0% in December. The clear trend from edge analysis suggested that the translucent bands started to form once a year near the end of winter and during the spring.

### 3.3. Age Determination Methods and Growth Parameters Estimation

For *P. macrocephalus*, mean ± sd of MSE from the k-folds cross-validation runs for the three age determination methods was 1.722 ± 0.010 for the integral method, 1.572 ± 0.048 for the quartile method, and 1.730 ± 0.011 for the back-calculation method. The quartile method has the lowest mean MSE compared to the other two methods and was considered the optimal method. A similar conclusion was obtained for *A. nibe* that the quartile method has the lowest mean MSE, with mean ± sd of MSE of 31.233 ± 5.005 for the integral method, 30.538 ± 4.011 for the quartile method, and 33.783 ± 1.760 for the back-calculation method. The quartile method was thus selected as the age determination method for both species.

For *P. macrocephalus*, the Richards model could not converge in the growth fitting analysis. The growth model that fits the length and age data best was VBGM with *AICc* differences Δ < 2 (indicating substantial support as the best model) [38] and Akaike weight *w* (the expected weight of evidence in favor of the model being the best among the four models) of 47% with sex combined (Figure 6, Table 2). The growth parameters of VBGM ($L_\infty$, $k$, $t_0$) were: 18.0 cm TL, 0.789 year$^{-1}$, and −0.872 year for sex-combined; 18.673 cm, 0.495 year$^{-1}$, −2.095 year for female; and, 17.864 cm, 0.892 year$^{-1}$, −0.571 year for male.

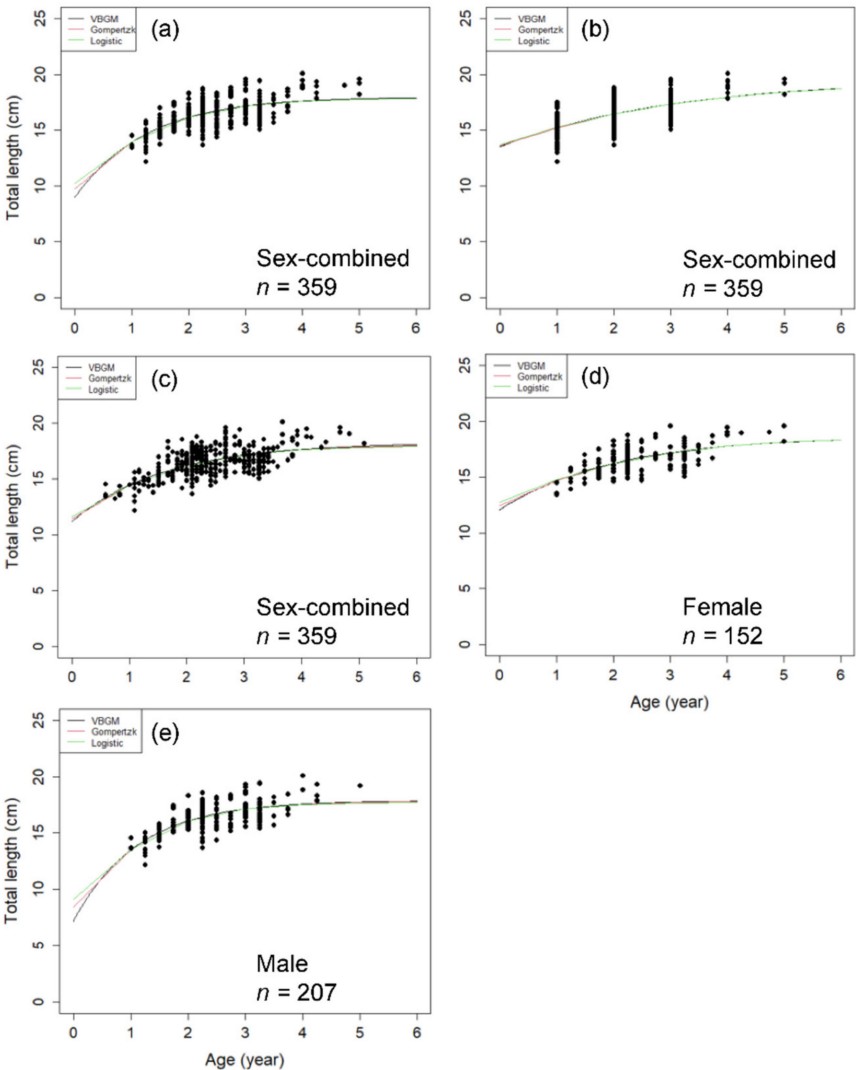

**Figure 6.** Age and length of sex-combined and sex-separated data and fitted growth curves of the three growth models for *P. macrocephalus* off southwestern Taiwan. (**a**,**d**,**e**) are the quartile of the marginal growth band method; (**b**) is the integral counting method; (**c**) is the back-calculation method.

**Table 2.** Parameter estimates (±standard error) from four candidate growth models, by sex, for *P. macrocephalus* and *A. nibe* off Southwestern Taiwan. The best-fit models are shown in bold.

| Species | Model | Sex | $L_\infty$ (cm TL) | $k$ (Year$^{-1}$) | $t_0$ (Year) | $p$ | AICc | ΔAICc | $w$ |
|---|---|---|---|---|---|---|---|---|---|
| *P. macrocephalus* | VBGM | C | 18.004 (0.315) | 0.789 (0.157) | −0.872 (0.368) | - | 1029.04 | 0 | 0.468 |
| *P. macrocephalus* | VBGM | F | 18.673 (0.982) | 0.495 (0.239) | −2.095 (1.280) | - | 445.08 | 0 | 0.386 |
| *P. macrocephalus* | VBGM | M | 17.864 (0.352) | 0.892 (0.202) | −0.571 (0.367) | - | 585.84 | 0 | 0.390 |
| *P. macrocephalus* | Gompertz | C | 17.921 (0.285) | 0.879 (0.162) | −0.564 (0.294) | - | 1029.81 | 0.774 | 0.318 |
| *P. macrocephalus* | Gompertz | F | 18.647 (0.955) | 0.532 (0.243) | −1.693 (1.015) | - | 445.39 | 0.311 | 0.330 |
| *P. macrocephalus* | Gompertz | M | 17.777 (0.316) | 1.002 (0.210) | −0.286 (0.290) | - | 586.17 | 0.328 | 0.331 |
| *P. macrocephalus* | Logistic | C | 17.850 (0.262) | 0.970 (0.168) | −0.311 (0.241) | - | 1030.59 | 1.556 | 0.215 |
| *P. macrocephalus* | Logistic | F | 18.643 (0.947) | 0.565 (0.246) | −1.354 (0.808) | - | 445.69 | 0.612 | 0.284 |
| *P. macrocephalus* | Logistic | M | 17.704 (0.288) | 1.112 (0.219) | −0.055 (0.236) | - | 586.52 | 0.678 | 0.278 |
| *A. nibe* | VBGM | C | 68.149 (8.182) | 0.130 (0.033) | −0.919 (0.413) | - | 2373.40 | 6.380 | 0.029 |
| *A. nibe* | VBGM | F | 60.548 (5.835) | 0.186 (0.047) | −0.266 (0.439) | - | 1443.83 | 4.090 | 0.075 |
| *A. nibe* | VBGM | M | 90.094 (37.291) | 0.070 (0.046) | −1.926 (0.870) | - | 911.61 | 1.736 | 0.180 |
| *A. nibe* | Gompertz | C | 57.162 (3.512) | 0.263 (0.036) | 1.919 (0.194) | - | 2369.53 | 2.514 | 0.202 |
| *A. nibe* | Gompertz | F | 54.732 (3.226) | 0.318 (0.052) | 1.853 (0.160) | - | 1441.39 | 1.648 | 0.255 |
| *A. nibe* | Gompertz | M | 62.594 (9.428) | 0.197 (0.050) | 2.313 (0.720) | - | 910.56 | 0.679 | 0.306 |
| *A. nibe* | Logistic | C | 53.111 (2.294) | 0.394 (0.040) | 2.958 (0.219) | - | 2367.02 | 0 | 0.709 |
| *A. nibe* | Logistic | F | 52.142 (2.306) | 0.448 (0.052) | 2.804 (0.160) | - | 1439.74 | 0 | 0. 582 |
| *A. nibe* | Logistic | M | 55.313 (5.285) | 0.322 (0.054) | 3.372 (0.615) | - | 909.88 | 0 | 0.429 |
| *A. nibe* | Richards | C | 55.345 (2.904) | 0.282 (0.035) | 1.843 (0.162) | $-3.661 \times 10^6$ ($3.477 \times 10^6$) | 2371.95 | 4.927 | 0.060 |
| *A. nibe* | Richards | F | 54.349 (3.260) | 0.322 (0.053) | 1.843 (0.183) | $-1.067 \times 10^5$ ($3.060 \times 10^6$) | 1443.53 | 3.788 | 0.088 |
| *A. nibe* | Richards | M | 57.794 (6.277) | 0.224 (0.049) | 1.944 (0.409) | $-6.994 \times 10^6$ ($6.283 \times 10^6$) | 913.12 | 3.248 | 0.085 |

Note: For sex: M, male; F, female; C, sex-combined.

For *A. nibe*, the best growth model was logistic with *AICc* differences Δ < 2 and Akaike weight *w* of 75% with sex combined (Figure 7, Table 2). The growth parameters of logistic ($L_\infty$, $k$, $t_0$) were: 53.11 cm, 0.39 year$^{-1}$, and 2.96 year for sex-combined; 52.14 cm, 0.45 year$^{-1}$, 2.80 year for female; and, 55.31 cm, 0.32 year$^{-1}$, 3.37 year for male.

The selection of the best model for each species was based on the quality of statistical fit assuming that the data could represent the potential species-specific growth pattern. Table 2 provides estimates of all the tested models for optional choices when considering the representativeness of the data set and the theoretical growth pattern from biological details of the fish [38,100].

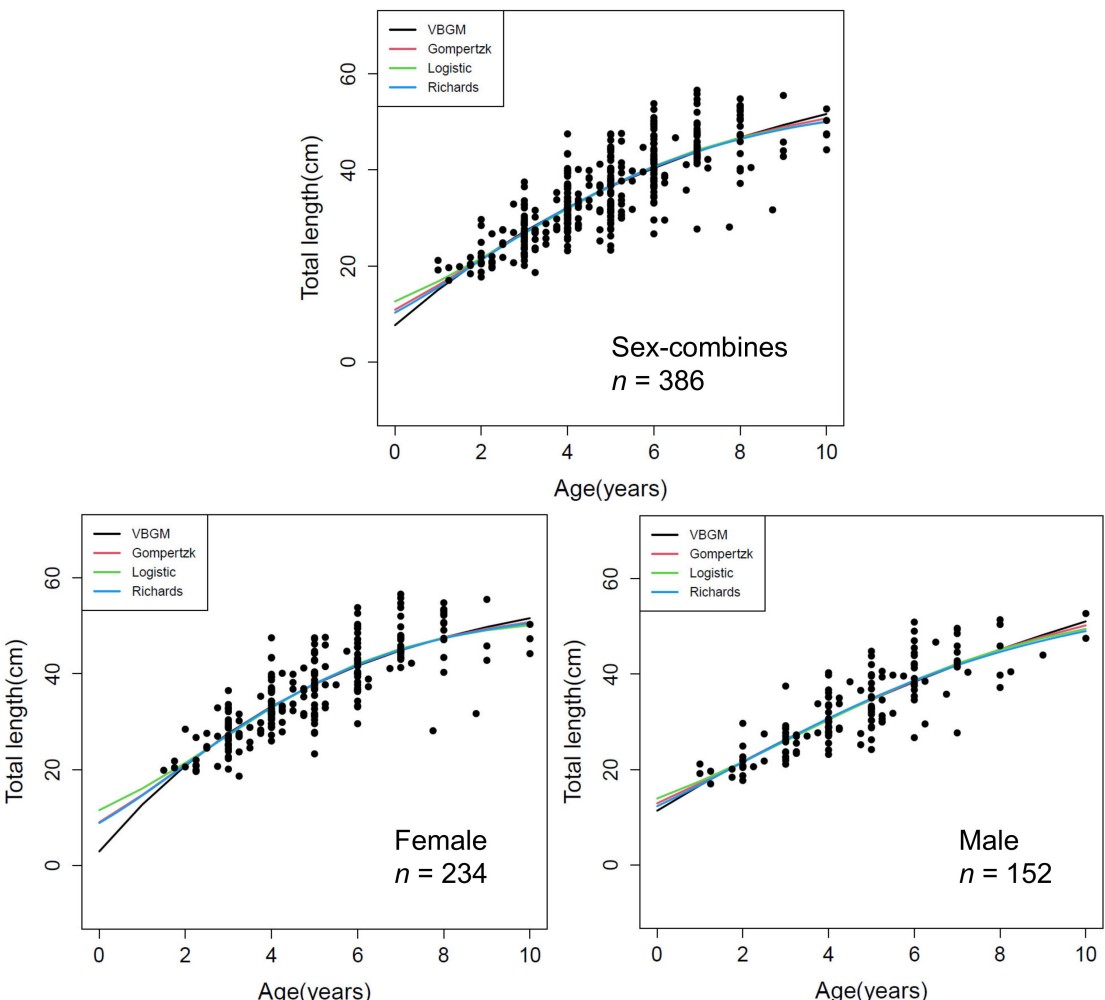

**Figure 7.** Age and length of sex-combined and sex-separated data and fitted growth curves of the three growth models for *A. nibe* off southwestern Taiwan.

*3.4. Global Estimates Compilation*

Few studies on *P. macrocephalus* and *A. nibe* were found in the literature. Especially for *A. nibe*, many were reported in grey literature and were cited in academic publications without detailed information (e.g., many were cited in Pauly [92], and information on the data sources are not available online). Due to the lack of studies in the two targeted species, studies from the same genus documented previously were also included in the analyses. A total of 62 growth records were obtained covering six species from ten regions with three estimation methods, including otolith, scale and length frequency (Table 1).

For *Pennahia* species, the age ranged from 0 to 10 years, and the growth parameters of VBGM ranged from 15.15–45.20 cm TL for $L_\infty$ and 0.181–1.94 year$^{-1}$ for $k$. The growth performance index was 2.10 to 3.16 (mean = 2.61), and for *P. macrocephalus* was between 2.10 to 2.77 (mean = 2.43). For *Atrobucca* species, the age ranged from 1 to 13 years, and the growth parameters ranged from 37.10–68.15 cm TL for $L_\infty$ and 0.116–0.523 year$^{-1}$ for $k$. The growth performance index was 2.58–3.05 (mean = 2.75), and for *A. nibe* was in the same range (mean = 2.76).

The two variables, $\log(L_\infty)$ and $\log(k)$, of the growth performance index formula and the results of the growth performance index, were shown in the auximetric plots (Figure 8). The same species tend to be in the same cluster with similar $\varphi$.

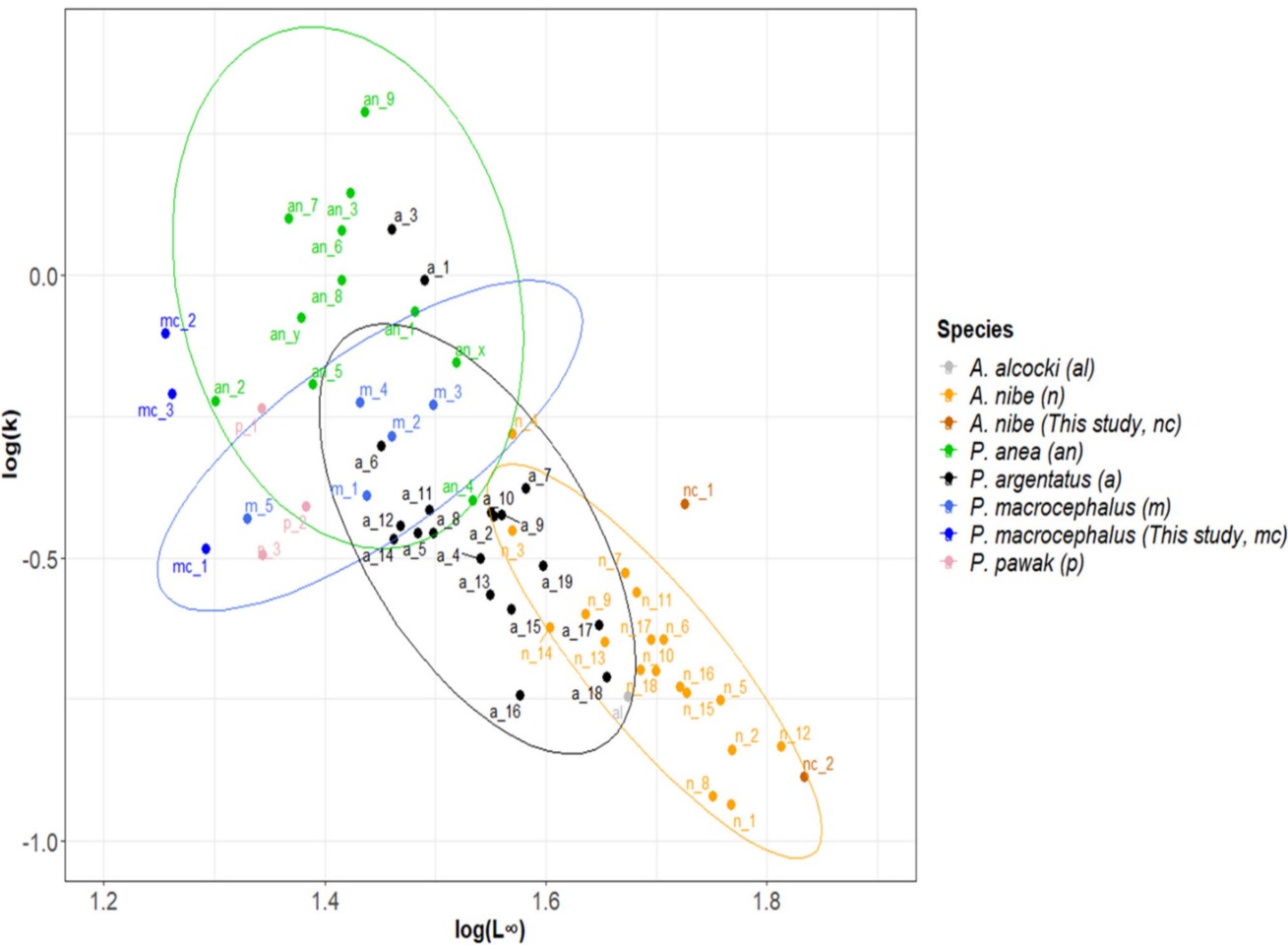

**Figure 8.** Auximetric plot of the two variables, $\log(L_\infty)$ and $\log(k)$, of the growth performance index formula obtained from the previous studies globally of Sciaenidae family. Ellipses with a 95% confidence interval. Code referred to Table 1.

## 4. Discussion

### 4.1. Length-Weight Relationships

In order to compare results from this study with past studies, the LWR in TL was transformed to a LWR in SL for *P. macrocephalus* using the length-length relationship [70]. Divergencies were noted among the LWR estimates in Table 3. Although the reasons for the divergencies have not been fully explored, in addition to possible sampling errors in data collection methods [101], the following differences in data handling may, in part, account for the observed differences. (1) Different logarithmic functions are used for the LWR parameter estimation. Both base-10 logarithmic ($\text{Log}_{10}$) [54,70] and natural logarithmic (Ln) functions [59,102–104] have been used to linearize the LWR equation for regression estimation, and most of the studies did not specify the logarithmic function they used. The two functions will result in different parameter values while the *b* parameter remains the same. (2) Uneven distribution of the samples by fish size, such as oversampling of medium-sized fish, can affect the regression fitness. For example, large fish were less fitted in the LWR of Tasi (Figure 19 in reference [34]) (Table 3), which might cause an overestimation of the allometric coefficient (*b* value). (3) Uneven distribution of sampling size among months, e.g., over-weighted samples were from the months when the fish grow faster or slower, could cause estimation differences since LWR may change seasonally with different levels of gonad maturity and other parameters [105]. Nhuận and Khuong [73] (Table 3) showed that *b* values estimated in the breeding season (third season) tended to be higher than in

other seasons. For this study, most of the samples were from the breeding season, and thus, the *b* value was likely overestimated to some extent.

**Table 3.** The coefficients *a* and *b* of the length–weight relationship equation (LWR) of *P. macrocephalus* (standard length) and *A. nibe* (total length) of the current and previous studies.

| Species | Region | a | b | Reference |
|---|---|---|---|---|
| *P. macrocephalus* | Beibu Gulf, SCS | $2.37 \times 10^{-5}$ | 3.248 | [29] |
| *P. macrocephalus* | Beibu Gulf, SCS | $2.16 \times 10^{-2}$ | 3.032 | [70] |
| *P. macrocephalus* | Beibu Gulf, SCS | $0.01 \times 10^{-6}{\sim}0.01 \times 10^{-7}$ | 2.95~3.57 | [73] |
| *P. macrocephalus* | Yunlin, TW | $4.30 \times 10^{-2}$ | 2.747 | [32] |
| *P. macrocephalus* | Southwest, TW ♀ | $4.70 \times 10^{-2}$ | 2.730 | [106] |
| *P. macrocephalus* | Southwest, TW ♂ | $4.08 \times 10^{-2}$ | 2.786 | [106] |
| *P. macrocephalus* | Southwest, TW | $9.90 \times 10^{-3}$ | 3.337 | [34] |
| *P. macrocephalus* | Southwest, TW | $2.77 \times 10^{-2}$ | 2.937 | This study |
| *A. nibe* | Yellow Sea and ECS | $2.03 \times 10^{-3}$ | 3.463 | [107]; cited in [108] |
| *A. nibe* | ECS | $2.84 \times 10^{-3}$ | 3.220 | [94]; cited in [108] |
| *A. nibe* | North, TW | $6.39 \times 10^{-4}$ | 3.207 | [109]; cited in [108] |
| *A. nibe* | Northeast, TW | $1.41 \times 10^{-3}$ | 2.934 | [98]; cited in [108] |
| *A. nibe* | Northeast, TW | $6.92 \times 10^{-3}$ | 3.100 | [96] |
| *A. nibe* | Southwest, TW | $1.21 \times 10^{-3}$ | 2.968 | [97] |
| *A. nibe* | Southwest, TW | $2.00 \times 10^{-3}$ | 2.907 | [110] |
| *A. nibe* | Oman Sea ♀ | $1.21 \times 10^{-2}$ | 2.939 | [95] |
| *A. nibe* | Oman Sea ♂ | $7.5 \times 10^{-3}$ | 3.074 | [95] |
| *A. nibe* | Southwest, TW | $1.45 \times 10^{-2}$ | 2.886 | This study |

Note: For Region: SCS, the South China Sea; TW, Taiwan; ECS, the East China Sea. ♀: female, ♂ male.

For *A. nibe*, differences in LWR were also observed (Table 3), which might result from several possible reasons, including seasonal or environmental changes, as proposed in Schneider [111] and De Giosa [112] and the abovementioned data handling issues. Sample distributions by fish size and month were evenly distributed in this study, so the LWR estimation was considered representative.

*4.2. Periodicity of Increment Formation—Edge Analysis*

Both edge analysis and marginal increment analysis (MIA) are commonly used methods to verify the increment periodicity of the otolith [20,21], with edge analysis being a qualitative approach and MIA a quantitative approach [40]. While MIA is generally considered more robust than edge analysis, edge analysis (or marginal analysis) only requires recording the otolith margin type, thus is relatively affordable both in terms of equipment and time [26] and was chosen in this study to examine the periodicity of increment formation. For *P. macrocephalus*, the occurrence of the translucent bands was lowest in February, increased in March and April, and then jumped to a high level from June to October (Figure 5), suggesting that the translucent bands were formed once a year near the end of winter and the early spring. This pattern was supported by an additional MIA (Supplementary Figure S2), which showed a continuously increasing trend of marginal increment from February to October.

The spawning season of this fish is from April to October (also the main fishing season), and the gonadosomatic index (GSI) value declines significantly from September to October [113]. The fish migrate to Taiwan mainly for spawning, and many large spawning fish species inhabit deeper depths after spawning (e.g., white-mouth croaker [114]) and thus are more difficult to catch by fisheries. Though sampling could not be performed year-round, this study assumed that most fish were hatched before or in October and that the lowest proportion of the translucent bands might be in the month without fish samples (November to January). The MIA of Attaqi [32] also showed a single sinusoidal cycle in which the highest appeared from August to September and decreased sharply in October. Results from this study and Attaqi [32] both support ring marks being formed once a year.

On the other hand, consecutive monthly samples were available for *A. nibe,* and the edge analysis showed a continuous increase in the occurrence of the translucent band from the lowest month in December to the highest month in September (Figure 5). The single sinusoidal cycle pattern also supported the annual formation of the ring mark.

The edge analysis and MIA are valuable tools for the validation of age estimates, but both have their limitations (refer to Campana [13] for a detailed discussion on the topic). Therefore, although the current analyses supported annual formation of the ring mark for the two species, further analyses are still needed for definite validations; parameters that would possibly affect ring formation (e.g., fluctuations of environmental factors and timing of high feeding sources) are also worth further investigation.

### 4.3. Age Determination Methods

A growth ring is a part of an otolith that consists of one translucent zone and one opaque zone [115]. When the annual formation of the growth ring is validated, the age is usually determined by the number of ring counts [20,116,117]. However, this method ignores valuable information contained within a year's growth. Although age estimates for older fish may be best approximated by integer ages because the edge proportion is usually hard to identify for older fish (as seen in Figure 7), information on decimal ages can be gathered in younger fish. The quartile method could retain the lost information and be almost as fast as the integral method when conducting. Cross-validation simulation demonstrated that the quartile method could provide more appropriate estimates of age than the integral method.

The back-calculation method has been applied in many studies [32,33]. It needs an assumption of the birthdate of the fish. Vanderkooy [33] described the age from this method as the biological age, derived from the information of capture month (sampling month), number of annuli, and accepted birthdate estimate. This method could provide better distribution of age estimates when developing a growth model than the integral method if the study has representative samples of the sizes closest to the breeding and juvenile stages to determine the correct birthdate. In this regard, this method needs beforehand studies on fish reproduction or daily-rings counts to provide birthdate information. For *P. macrocephalus*, Attaqi [32] set 1st August as the hypothesized hatching date based on Wang [118], but the study from He [113] suggested that the spawning season could be as long as seven months since April. In this case, the difficulty in determining the birthdate would cause uncertainty in the estimate of age. Comparatively, the quartile method is more straightforward in interpreting the age of fish otoliths with both efficiency and accuracy.

Only Tsai [34] estimated the growth parameters of *P. macrocephalus* based on "daily rings", while others used annuli. The study suggested one ring formed every two days ("two-day ring") in the otolith based on the comparisons of the back-calculated birthdate and the GSI trend, and so the maximum age of this species was estimated to be 2.2 years. Besides lacking validations of the periodicity of increment formation and the first increment [27], the study also needs to consider the possibility that nondaily deposition can occur in suboptimal or extreme environmental conditions [119]. With these issues being addressed, daily rings for growth studies could provide more accurate estimates than the three methods described here, although the high cost of time and human resources need to be considered.

### 4.4. Growth Parameter Estimates—Global and Local

The global estimates of growth parameters for the Sciaenidae genera, together with those of this study, were collated and plotted for an overview (Figure 8) for the first time. The overall trend is downwards, from *P. anea* in the top-right corner to *P. macrocephalus*, *P. pawak*, *P. argentata*, and *A. nibe* in the bottom-left corner with comparatively higher $L_\infty$ and smaller $k$.

Estimates of each species were clustered as a group, demonstrating that each species has a relatively similar growth performance [44], which is also noted for tuna species [43,45]. The elongated clusters were also found in catfish (Ariidae) and sardines (Clupeidae), despite the different sampling regions or aging methods [41,42]. Estimates of the same species but outside the group might have specific reasons or need further investigation [27,47,48]. For example, estimates of a_1 and a_3 of *P. argentata* are far away from the cluster of the species, and further examining the estimates could find that the two estimates were specifically on smaller fish (SL $\leq$ 26.59 cm for a_1 and SL $\leq$ 24.47 cm for a_3) that have smaller $L_\infty$ but much higher $k$. As demonstrated in other studies, growth parameter estimates would be significantly affected by the lack of very young or old individuals [120], and the lack of older fish might underestimate asymptotic length [121].

The type of aging structure has been observed to significantly affect the estimation results [27,48]. However, although otolith, scale or length-frequency data were used for growth studies on the species in Table 1, differences in the estimates using different structures were not obvious for unknown reasons.

There were not many growth studies for *P. macrocephalus*. As explained earlier, Tsai's [34] study (m_4) was likely biased. Excluding this estimate, the estimates from the South China Sea (m_1 to m_3) departed from those from Taiwan waters (m_5 and mc_1–mc_3), suggesting possible regional effects (different environmental productivity or fishing pressure) on the estimation (Figure 8). Diverse results using different age determination methods in this study were observed in the figure. The recommended estimate with higher precision (using quartile method, mc_2) departs from Attaqi's [32] estimate that studied the fish in a similar location with smaller $L_\infty$ and higher $k$. In addition to many possible technical reasons in the estimation [27], the phenomenon might affect high fishing pressure. The data used in Attaqi [32] was from 2010–2011, about a decade before the data used for this study (2018–2021). Total landings of the category of "white mouth croaker" (*P. macrocephalus* composed the majority of the catch) in the market had declined drastically from 800 tons in 2011 to <200 tons after 2018 [122]. High fishing pressure can lead to less competition for food and space in the fish population and thus produce higher growth coefficients [7–9,123]. On the other hand, high fishing pressure can cause the loss of larger individuals and a decrease of relative abundance of target species which have the capacity to grow to large size, and thus is attributed to a smaller estimate of asymptotic length [7,124]. The same inference was drawn in Yi [80] for *P. pawak* in which high fishing pressure contributed to the decline of $L_\infty$ from 24.15 cm in 2008–2009 to 22.05 cm in 2018–2019, a decade later. These findings suggested the necessity to establish a management framework on the species to reduce fishing pressures.

For *A. nibe*, the growth estimates of VBGM (nc_2) from this study and from the previous studies (n_1 to n_18) formed a cluster, and all were within the 95% confidence interval. The best estimate in this study was the logistic model (nc_1); however, this estimate deviated from the 95% confidence interval ellipse, which was likely due to the difference in growth models. Except for one of the estimates from a study in the Oman Sea (n_10), the rest were all from the northeast Pacific Ocean (Taiwan, Japan, Korea). However, since this species is not a highly migrating species, those fish in different waters might belong to different stocks, so the growth parameters were not comparable.

Otolith, scale and length-frequency data have been used to estimate the parameters. Estimates using length-frequency data tend to have lower $k$ than estimates using hard structures for dolphinfish and flyingfishes [27,48]; however, this pattern has not been observed for *A. nibe.* Compared to the estimates from Taiwan using the same aging material of otolith (n_11, n_12, n_15–n_18), the $k$ estimates of this study were much higher. The obvious difference between these studies was that the data of previous studies were conducted about a decade ago (1993 and 2008). Based on market data, the annual landing of the species has dropped 75% from the early 2010s to the late 2020s and hence fishing pressure might have contributed to the increase in $k$ estimates. The other difference was that this study included more samples of older fish (ages 9 and 10), which could make

the estimates of $L_\infty$ higher and $k$ lower [121]. However, care needs to be taken that the quality of this study might be affected by the lack of small fish samples (age 1), as suggested in the following.

Although the VBGM was selected as the best model for *P. macrocephalus* because of the lowest $\Delta AICc$, the differences among models were not substantial (Table 2). The growth curves at Figure 6 also showed little differences in fit among the models where data are present; the differences occur mainly beyond the data (e.g., age 0). In addition, all models underestimate lengths of the oldest ages (4+). For *A. nibe*, the logistic model fits the age groups 2–8 better but diverted in age 0–1 and age 9+. Meanwhile, $L_\infty$ estimates for both species fall beyond the range of observed data. These indicated the limitation of the lack of old and very young (0–1) age group samples in this study. To refine the growth estimates (especially the $L_\infty$ estimates), further sample collections targeted at larger fish and very young (age 0–1) fish are needed in the future.

## 5. Conclusions

*P. macrocephalus* and *A. nibe* are among the most abundant commercial Sciaenidae species in southwestern Taiwan. Landings of the two species have dropped rapidly since 1998, presumably due to overfishing. Most studies on the estimation of growth parameters for the two species were conducted a decade ago, and the majority of them were published in the grey literature or in Chinese. To fill the information gap, this study provides reliable growth estimates and length–weight relationships (in Section 3.1 and Table 2) as updated information to the global literature and provides an overview of growth parameters for the two important croakers (in Table 1), which could be used for the development of a sound stock assessment on the resources. Comparison of the resulted growth parameters with those estimated a decade ago suggested that these two species have undergone high fishing pressures and induced consequent higher growth coefficients and smaller asymptotic length, and thus suggested the need for establishing a management framework to reduce fishing pressures.

Methodology analyses of this study suggested that the quartile of marginal growth band method could provide more appropriate estimates of age from otoliths than the other two methods. The results suggested the VBGM as the optimal growth model for *P. macrocephalus* and the logistic model for *A. nibe*, based on the samples collected. However, both results need further refinements by collecting more samples of large fish and very young fish. Finally, the auximetric plot of this study supported the argument that the same fish species has a similar growth performance and could be used as a quick tool for identifying outliers of growth estimates for further investigations.

**Supplementary Materials:** The following supporting information can be downloaded at: https://www.mdpi.com/article/10.3390/fishes7050281/s1, Table S1: Detail information on the estimates of growth parameters for *Pennahia* and *Atrobucca* species from studies published globally; Table S2: Length and weight raw data for *Pennahia macrocephalus*; Table S3: Length and weight raw data for *Atrobucca nibe*; Figure S1: Images of otoliths (sagitta, lapillus, and asteriscus) of *P. macrocephalus* and *A. nibe*; Figure S2: The marginal increment ratio (MIR) of *P. macrocephalus.*

**Author Contributions:** Conceptualization, S.-C.H. and S.-K.C.; Data curation, S.-C.H. and J.-S.H.; Formal analysis, S.-C.H. and T.-L.Y.; Funding acquisition, S.-K.C. and J.-S.W.; Investigation, S.-C.H. and C.-C.L.; Methodology, S.-C.H., S.-K.C. and T.-L.Y.; Project administration, S.-K.C. and C.-C.L.; Resources, S.-K.C.; Software, S.-C.H. and T.-L.Y.; Supervision, S.-K.C. and J.-S.W.; Validation, J.-S.H.; Visualization, S.-C.H.; Writing—original draft, S.-C.H.; Writing—review & editing, S.-K.C. and C.-C.L. All authors have read and agreed to the published version of the manuscript.

**Funding:** This research was funded by the Ministry of Science and Technology (MOST), Taiwan, grant number MOST 106-2611-M-110-006, MOST 108-2611-M-110-002, MOST 109-2611-M-110-004.

**Institutional Review Board Statement:** Not applicable.

**Data Availability Statement:** Data of this study is included in the report, the Supplementary Materials or may be made available by the authors.

**Acknowledgments:** The authors appreciate the technical support from Ying-Chu Chen in the otolith ring reading and the financial support of the Ministry of Science and Technology for the open access publication fee.

**Conflicts of Interest:** The authors declare no conflict of interest.

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
