# Peer review of "Length–Weight Relationships, Growth Models of Two Croakers (Pennahia macrocephalus and Atrobucca nibe) off Taiwan and Growth Performance Indices of Related Species"

_fishes, doi:10.3390/fishes7050281_

Round 1
Reviewer 1 Report
The manuscript contains essential information on the age and growth of two croakers from Taiwan. The manuscript presents a sound study with a simple experimental design. I have provided some minor comments as they arise in the manuscript.
1. Line 48: The authors should also mention other ageing structures.
2. Line 63: MIA is a more robust method than EA. The authors should glorify the utility of MIA.
3. Line 147: Replace in with for
4. Line 148: What was the ratio of the Epoxy Resin and Epoxy Hardener?
5. Line 150: Sandpaper or CarbiMet Abrasive?
6. Line 157: The authors have not added any information on abandoned results. Please add this information.
7. Line 333: The quality of Fig. 7 should be enhanced. The thickness of growth curve lines should be increased.
Reviewer 2 Report
This is a very interesting methodological study on the age and growth of important fisheries fish species. Considering that both age structure and growth parameters are essential to modern stock assessment and development of management measures the paper is highly relevant.
As a recommendation, I would like to ask the authors to tell something about the ecology of the species and the number of annual catches at Lines 104-105.
Also, It will be good to add the hypothesis of your study. Besides you need to specify your expectations with relevant references.
The absence of a conclusion at the end of the paper is a bit disappointing. In the current form, the paper looks as not finished.
Reviewer 3 Report
It is a well-written and structured article with good methodological support and interesting results, however, it requires small adjustments to improve its quality. Suggestions associated with the line number in the manuscript are described below for easy identification.
It is suggested to review the scale in figure 1. Apparently the size of the two species is the same, however the data presented shows that A. nibe is much larger
The paragraph between lines 106 to 115 is out of place as it is methodology. It is suggested to delete. Instead it could look like:
….Taiwan (Figure 1). This study provides reliable growth estimates…. and adds updated information ….croakers.
The sampling has a bias as it comes from the market, that is, commercial sizes. However, this flaw is recognized in the discussion.
Lines 126 and 128 the term “southwestern Taiwan” is repeated. Please correct
Line 136. I suggest adding the text: "the market samples used in this study come from these fishing areas".
Line 139. Missing to include the meaning of BW and TL
Line 141-142. Because ANOVA?, the contrast is between two means, therefore the correct statistic to use is t-student. Although the results appear in the ANOVA of the regression, it does not mean that an ANOVA was done. Correct same error on lines: 258-259, 274-275
Delete text from line 235 to 236, as the same text appears in line 237
Line 269. The lines and circles. Add circles
Line 301-302. It is questionable because we use the same quartile method for both species. Given the different behavior of the two species, I consider that the analysis should also be carried out for A. nibe, since it is possible that in this case, one of the other methods turns out to be better.
Sections 3.3. and 3.4 should be integrated into one to avoid repeating information
Lines 359 to 361 are results, not discusión, so this should be passed to that section including table 4.
Line 371. Fig. 19 of 34. does not exist delete
Line 386 Include space in abovementioned. Above mentioned
Line 402 GSI Write Complete. Gonadosomatic Index.
Line 419 To validate the age, it is important to identify the factor causing the formation of growth rings. I suggest complementing the validation with information regarding what is considered to be the factors that produce the formation of marks in the otoliths, that is, annual fluctuations in environmental factors, times of high and low feeding sources, reproductive reasons, etc.
I share an article that may be useful to identify how to perform age validation taking into account the factors that cause the formation of growth rings.
Line 435. And when you have representative samples of the sizes closest to the breeding and juvenile stages
Line 441-442. …..efficiency and precision, but it is not enough
Line 488 to 490. Sounds contradictory. If it has the capacity to grow to a larger size, then an larger asymptotic length would be expected, not smaller as stated here. Clarify.
A comparison of individual values of the growth parameters Linf and K is not valid since these vary inversely proportionally. Therefore, it is better to compare the overlap of the growth curves generated by each pair of growth parameters (Linf, K), especially when comparing the parameters estimated in this study against the estimates of other authors.
There is an excessive number of bibliographic references (124), I recommend reviewing and citing only those that provide relevant information for the manuscript
